# A Benchmark Suite for Evaluating Neural Mutual Information Estimators on Unstructured Datasets

**Kyungeun Lee**
LG AI Research
Seoul, Republic of Korea
kyungeun.lee@lgresearch.ai

**Wonjong Rhee**
Seoul National University
Seoul, Republic of Korea
wrhee@snu.ac.kr

## Abstract

Mutual Information (MI) is a fundamental metric for quantifying dependency between two random variables. When we can access only the samples, but not the underlying distribution functions, we can evaluate MI using sample-based estimators. Assessment of such MI estimators, however, has almost always relied on analytical datasets including Gaussian multivariates. Such datasets allow analytical calculations of the true MI values, but they are limited in that they do not reflect the complexities of real-world datasets. This study introduces a comprehensive benchmark suite for evaluating neural MI estimators on unstructured datasets, specifically focusing on images and texts. By leveraging same-class sampling for positive pairing and introducing a binary symmetric channel trick, we show that we can accurately manipulate true MI values of real-world datasets. Using the benchmark suite, we investigate seven challenging scenarios, shedding light on the reliability of neural MI estimators for unstructured datasets.

## 1 Introduction

Mutual Information (MI), denoted as $I(X;Y)$, serves as a fundamental measure in quantifying the dependency between two random variables [Cover, 1999]. It is mathematically defined as:

$$I(X;Y) = \mathbb{E}_{p(x,y)} \log \left[ \frac{p(x,y)}{p(x)p(y)} \right].$$

In practice, we often rely on the estimations instead of the exact calculation of MI because we can only access the examples sampled from joint and marginals but not the underlying distribution functions ($p(x,y)$ and $p(x)p(y)$). To this end, various sample-based MI estimators have been proposed [Fraser and Swinney, 1986, Shwartz-Ziv and Tishby, 2017, Kraskov et al., 2004, Belghazi et al., 2018, Poole et al., 2019, Song and Ermon, 2019, 2020, Cheng et al., 2020], and they have played a key role in improving the deep learning performance across diverse applications, including generative models [Chen et al., 2016], language representation learning [Oord et al., 2018, Wang et al., 2020], domain generalization [Li et al., 2022], anomaly detection [Lei et al., 2023], and self-supervised learning [Hjelm et al., 2018, Bachman et al., 2019, Chen et al., 2020, Chen and He, 2020, Grill et al., 2020].

Despite the huge success of MI estimators in developing useful real-world applications, the accuracy of MI estimators on real-world datasets largely remains underexplored because the true MI values cannot be calculated for such datasets. Gaussian datasets, the primary benchmark in the existing studies [Belghazi et al., 2018, Poole et al., 2019, Song and Ermon, 2019, 2020, Cheng et al., 2020], do not adequately represent the complexity of real-world datasets. This raises a fundamental question: *do estimators that perform well on Gaussian datasets also excel with more complex datasets like images or texts?* Recently, Czyż et al. [2023] have explored non-Gaussian datasets to evaluate MI

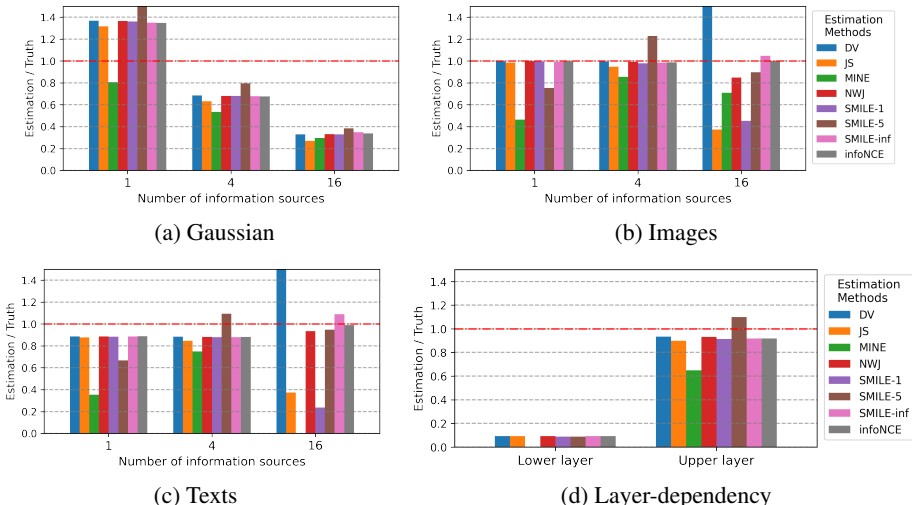

Figure 1: Exemplary findings from our benchmark suite. The ratio between estimated MI value and true MI value is shown in y-axis. A ratio close to 1.0 indicates a highly accurate MI estimation. (a, b, c) When the number of independent information sources is increased to 16, all MI estimators become inaccurate for Gaussian dataset (shown in (a)) but some MI estimators remain accurate for image dataset (shown in (b)) and text dataset (shown in (c)). (d) When MI is estimated using embeddings from different layers of ResNet-50, MI estimators stay accurate only for the upper layers.

estimator accuracy. However, their evaluation is also limited to datasets with tractable distributions, such as Student's t-distributions, still far from being representative of real-world datasets. To address this gap, we present a method for evaluating MI estimators on any dataset in the absence of underlying distribution functions. Our approach employs same-class sampling as positive pairing [Lee et al., 2023] and binary symmetric channels [Cover, 1999] for precise manipulation of the true MI values.

We have developed a benchmark suite based on our method, encompassing three data domains for Gaussian multivariates, images, and sentence embeddings. To demonstrate its usefulness, we examine performance of several neural MI estimators over seven key aspects in Section 5. The seven aspects are critic architecture, critic capacity, choice of neural MI estimator, number of information sources, representation dimension, strength of nuisance, and layer-dependency. Through the examination of the seven aspects, we report interesting findings from our benchmark suite. For instance, when the number of information sources become large, MI estimation is relatively robust for image and text datasets than for Gaussian dataset as shown in Figure 1 (a,b,c). We also report that a larger critic capacity does not ensure a higher estimation accuracy, there is no single MI estimator that provides a universally superior performance over the three data domains, representation dimension can be as large as 10,000 without degrading the accuracy of MI estimators, MINE [Belghazi et al., 2018] turns out to be relatively robust to nuisance, and MI estimation can be surprisingly inaccurate for lower-layer representations while it is typically much more accurate for upper-layer representations (see Figure 1d). The benchmark suite and codebase are available in `https://github.com/kyungeun-lee/mibenchmark`.

## 2 Backgrounds: Neural Mutual Information Estimators

Mutual information between two random variables $X$ and $Y$ is defined as follows.

$$I(X;Y) \triangleq KL(p(x,y)||p(x)p(y)) = \mathbb{E}_{p(x,y)} \log \left[ \frac{p(x,y)}{p(x)p(y)} \right]$$

When only a finite set of joint samples is available, the exact MI cannot be calculated, but an estimation can be made. Among the known MI estimation methods, including simple binning [Fraser and Swinney, 1986, Shwartz-Ziv and Tishby, 2017] and non-parametric kernel-density estimators [Kraskov et al., 2004], variational estimators based on variational bounds and deep neural networks (DNN) modeling have become dominant for complex datasets [Belghazi et al., 2018, Poole et al., 2019, Song

Table 1: Summary of neural mutual information estimators. For the optimization step, we learn $f^*(x, y)$ by maximizing the optimization loss $\mathcal{L}(f(x, y))$ with a given batch size $K$. For the estimation step, we evaluate MI values as $\hat{I}(X; Y)$. DV, NWJ, and InfoNCE utilize the same formulation for optimization and estimation.

| Estimator | Optimization Loss - $\mathcal{L}(f(x, y))$ | Estimate Evaluation - $\hat{I}(X; Y)$ |
|---|---|---|
| DV [Donsker and Varadhan, 1983] | $\mathcal{L}_{\text{DV}}(f(x, y)) = \hat{I}_{\text{DV}}(X; Y) = \mathbb{E}_{p(x,y)}[f[x, y]] - \log \mathbb{E}_{p(x)p(y)}[e^{f(x,y)}]$ | |
| NWJ [Nguyen et al., 2010] | $\mathcal{L}_{\text{NWJ}}(f(x, y)) = \hat{I}_{\text{NWJ}}(X; Y) = \mathbb{E}_{p(x,y)}[f[x, y]] - e^{-1}\mathbb{E}_{p(x)p(y)}[e^{f(x,y)}]$ | |
| InfoNCE [Chen et al., 2018] | $\mathcal{L}_{\text{InfoNCE}}(f(x, y)) = \hat{I}_{\text{InfoNCE}}(X; Y) = \mathbb{E}_{p^K(x,y)}\left[\frac{1}{K}\Sigma_{i=1}^K \log \frac{f(x_i, y_i)}{\frac{1}{K}\Sigma_{j=1}^K f(x_i, y_j)}\right]$ | |
| JS [Poole et al., 2019] | $\mathbb{E}_{p(x,y)}\left[-\text{Softplus}(-f(x, y))\right] - \mathbb{E}_{p(x)p(y)}\left[\text{Softplus}(f(x, y))\right]$ | $\hat{I}_{\text{NWJ}}(X; Y)$ |
| MINE [Belghazi et al., 2018] | $\mathbb{E}_{p(x,y)}[f[x, y]] - \frac{\mathbb{E}_{p(x)p(y)}[e^{f(x,y)}]}{\text{ExponentialMovingAverage}(\mathbb{E}_{p(x)p(y)}[e^{f(x,y)}])}$ | $\hat{I}_{\text{DV}}(X; Y)$ |
| SMILE-$\tau$ [Song and Ermon, 2019] | $\mathcal{L}_{\text{JS}}(f(x, y))$ | $\mathbb{E}_{p(x,y)}[f[x, y]] - \log \mathbb{E}_{p(x)p(y)}[\text{clip}(e^{f(x,y)}, e^{-\tau}, e^{\tau})]$ |

and Ermon, 2019, 2020, Cheng et al., 2020]. These DNN-based estimators are commonly referred to as neural MI estimators. In this study, we focus on neural MI estimators due to their superior performance in handling large sample sizes and high-dimensional data, as demonstrated in previous works [Gao et al., 2015, Belghazi et al., 2018, Poole et al., 2019, Czyż et al., 2023].

Variational MI estimators are based on two steps. First, an analytical bound is derived where the bound is based on a critic function $f(x, y)$. Second, the bound is made tight by optimizing for a supremum or an infimum over $f(x, y)$. In recent works, deep neural networks have been used to model the critic function. When a proper loss function is chosen and the learning of $f(x, y)$ is successful, the variational estimations have been shown to be accurate for Gaussian datasets [Belghazi et al., 2018, Poole et al., 2019, Song and Ermon, 2019, Czyż et al., 2023].

**Definition 2.1** (Variational MI estimators [Poole et al., 2019]). Let $X, Y$ be two random variables taking values in $\mathcal{X}, \mathcal{Y}$, and $\mathcal{D} = \{(x_i, y_i)\}_{i=1}^N \sim X, Y$ denote the set of samples drawn from a joint distribution over $\mathcal{X}$ and $\mathcal{Y}$. The variational bounds of $I(X; Y)$ are formulated as:

$$I(X; Y) \geq \hat{I}(X; Y) = 1 + \mathbb{E}_{p(x,y)}\left[\log \frac{e^{f(x,y)}}{a(y)}\right] - \mathbb{E}_{p(x)p(y)}\left[\frac{e^{f(x,y)}}{a(y)}\right]$$

where $a(y) > 0$ is any value or function of $y$. A variety of MI estimators are defined by adopting different $a(y)$. For example, $a(y) = e$ (constant) corresponding to $\hat{I}_{\text{NWJ}}(X; Y)$ [Nguyen et al., 2010] (also known as $f$-GAN KL [Nowozin et al., 2016] and MINE-$f$ [Belghazi et al., 2018]) and $a(y) = \frac{1}{K}\sum_{i=1}^K e^{f(x_i, y)}$ corresponding to $\hat{I}_{\text{InfoNCE}}(X; Y)$ [Oord et al., 2018].

For a neural MI estimator, a DNN is used to model the *critic* function $f(x, y)$, and there are two associated steps. The first is the optimization (or training) step where the DNN parameters are learned. The second is the estimation step where the actual MI values are inferred with the optimized DNN. Variational MI estimators, such as DV [Donsker and Varadhan, 1983], NWJ [Nguyen et al., 2010], and InfoNCE [Oord et al., 2018], use a single loss function for both optimization and estimation, and the loss function corresponds to the theoretical MI bound in use. Other variational MI estimators, such as JS [Nowozin et al., 2016], MINE [Belghazi et al., 2018], and SMILE [Song and Ermon, 2019], adopt small modifications in either optimization or estimation to improve the robustness or accuracy of the estimator. The most popular variational MI estimators are summarized in Table 1.

Common choices for the critic function $f(x, y)$ include (1) the inner product critic $f_{\text{inner}}(x_i, y_j) = x_i^T y_j$, (2) bilinear critic $f_{\text{bi}}(x_i, y_j) = x_i^T W y_j$ where $W$ is trainable, (3) separable critic $f_{\text{sep}}(x_i, y_j) = f_1(x_i)^T f_2(y_j)$, and (4) joint critic $f_{\text{joint}}(x_i, y_j) = f_3([x_i, y_j])$. Here $f_1$, $f_2$, and $f_3$ are typically shallow MLPs and they model the relationship between all pairs of $(x_i, y_j) \, \forall i, j \in [1, K]$.

## 3 Related Works

Despite its theoretical validity, MI estimators present a few disadvantages because we typically have access to samples, but not to the underlying distribution functions [Poole et al., 2019, Song and Ermon, 2019, Paninski, 2003, McAllester and Stratos, 2020]. Most estimators exhibit sub-optimal

performance, particularly when batch size $K$ is small and true MI is large. For instance, InfoNCE can result in a high bias because it is upper bounded by $\log K$ [Oord et al., 2018]. McAllester and Stratos [2020] noted that any distribution-free high-confidence lower bound on MI cannot be larger than $\mathcal{O}(\log K)$. In contrast, most estimators result in a variance that can increase exponentially with the increase in the true MI value [Poole et al., 2019, Song and Ermon, 2019, Xu et al., 2019]. While these findings are enlightening, the limitations of estimators have been assessed mostly for the Gaussian benchmarks only instead of for the real-world datasets.

In efforts to evaluate MI estimators beyond Gaussian datasets, Song and Ermon [2019] proposed self-consistency tests using MNIST and CIFAR-10 datasets. They found that all estimators were not accurate for images. However, the analysis can be highly misleading because they evaluated based on the approximated metrics instead of the true MI. Czyż et al. [2023] suggested a non-Gaussian benchmark with tractable distributions (*e.g.*, multivariate student, uniform distribution) with the formulated mapping functions (*e.g.*, spiral transformation), enabling access to true MI. They considered non-Gaussian datasets that have tractable distribution functions, but the results are not representative of unstructured datasets, such as images and texts. In this study, we introduce a comprehensive method for evaluating MI estimators with no constraint on the type of data domains. In particular, we present a benchmark suite that allows assessment and manipulation of the true MI for images and sentence embeddings.

## 4    Proposed Method and Benchmark Suite

We propose a comprehensive method for evaluating neural MI estimators across various data domains. While our method is applicable to any choice of data domain, we focus on three types of data domains in our benchmark suite: (1) a multivariate Gaussian dataset ($\mathcal{D}_{\text{Gaussian}}$), corresponding to the most common case for evaluating MI estimators in the existing works [Poole et al., 2019, Song and Ermon, 2019, McAllester and Stratos, 2020]; (2) an image dataset consisting of digits ($\mathcal{D}_{\text{vision}}$), as an example of vision tasks; and (3) a sentence embedding dataset consisting of BERT embeddings of movie review datasets ($\mathcal{D}_{\text{NLP}}$), as an example of NLP tasks. We first consider the general formulation for Gaussian dataset, define three factors that can affect MI values, introduce same-class sampling, propose a method for generating unstructured datasets with adjustable true MI values, and finally explain how binary symmetric channel trick can be employed for manipulating true MI to any non-integer value.

### 4.1    General formulation for Gaussian dataset

Consider a dataset with $K$ pairs of samples, where $(x_i, y_i)$ is sampled from a joint distribution $p(x, y)$. An MI estimator utilizes the dataset as its input and evaluates the estimated mutual information $\hat{I}(X; Y)$. If the estimation is accurate, $\hat{I}(X; Y)$ should be close to the true mutual information $I(X; Y)$. In previous studies, a Gaussian dataset associated with a multivariate Gaussian model was utilized to assess neural MI estimators [Belghazi et al., 2018, Poole et al., 2019, Song and Ermon, 2019, 2020, Cheng et al., 2020]. The Gaussian dataset has Gaussian samples with zero mean and a component-wise correlation of $\rho$ between $X$ and $Y$. The true MI is known and can be expressed analytically as $I(X; Y) = -\frac{d_g}{2} \log\left(1 - \rho^2\right)$, where $x \in \mathbb{R}^{d_g}$ and $y \in \mathbb{R}^{d_g}$.

### 4.2    Definitions of $d_s$, $d_r$, and $Z$

In the benchmark suite, we design and focus on three essential factors that can affect mutual information $I(X; Y)$, especially for unstructured datasets. For random variables $X$ and $Y$ with a joint distribution $p(x, y)$, they can be defined as follows.

**Definition 4.1** (Number of information sources $d_s$). $d_s$ is the number of independent scalar random variables used to form the mutually shared information between $X$ and $Y$.

**Definition 4.2** (Representation dimension $d_r$). $d_r$ is the size of the observational data. When $X$ and $Y$ are of the same size, it is the length of the vector formed by flattening either $X$ or $Y$.

**Definition 4.3** (Nuisance $Z$). Nuisance to a random variable $X$ is defined as an equal-size random variable $Z$ sharing no information with $X$. Mathematically, $Z$ satisfies $I(X; Z) = 0$. Nuisance to $(X, Y)$ can be defined similarly where $Z$ is of the same size as $(X, Y)$ and $I(X, Y; Z) = 0$.

As an example, consider the Gaussian dataset. Its number of information sources $d_s$ is equal to $d_g$, its representation dimension $d_r$ is equal to $d_g$, and the dataset contains no nuisance. The effects of three factors on neural MI estimators will be analyzed in Section 5.

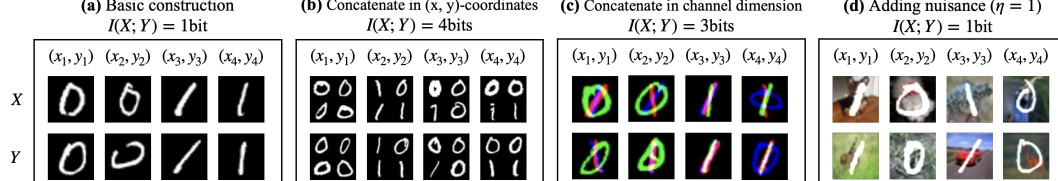

Figure 2: Four image examples of generating datasets with known true MI values. $X$ and $Y$ consist of random images drawn from the MNIST dataset. (a) Basic construction: only the images of class 0 or 1 are considered and $X$ and $Y$ are sampled to share the class information. By choosing $C$ to be either 0 or 1 with probability 0.5, $H(C)$ becomes 1 and therefore $I(X;Y) = H(C) = 1$. (b) Combining four independent images in 2-D to form a single image: $I(X;Y) = 4$ because $H(C) = H_{\text{left,up}} + H_{\text{left,down}} + H_{\text{right,up}} + H_{\text{right,down}} = 4$. (c) Combining three independent images in channel dimension to form a color image: $I(X;Y) = 3$ because $H(C) = H_{\text{green}} + H_{\text{red}} + H_{\text{blue}} = 3$. (d) Adding nuisance: an independently chosen background image from CIFAR-10 [Krizhevsky et al., 2009] is inserted as nuisance. Because the nuisance is independently chosen for $X$ and $Y$, they do not affect the true MI [Lee et al., 2023]. Therefore, $I(X;Y) = 1$.

### 4.3 Theoretical background: same-class sampling for positive pairing

Same-class sampling for positive pairing was proposed in Lee et al. [2023]. The key idea is to allow only the class information to be shared between two random variables $X$ and $Y$, such that the true MI can be proven to be the same as the entropy of class variable $C$, *i.e.*, $I(X;Y) = H(C)$. The proofs require mild assumptions of either a lower bound estimate of MI being equal to $H(C)$ or the existence of an error-free decoder (Theorem 3.1 and 3.2 in Lee et al. [2023]). In Lee et al. [2023], the first mild assumption was shown to be closely satisfied for commonly used image datasets, through extensive empirical evaluations. The second one essentially implies that the true MI is equal to $H(C)$ when the class information is easily decodable. An example is MNIST dataset whose digit information as the class label is known to be easily decodable. Similarly, for NLP datasets, sentence embeddings of the IMDB dataset [Maas et al., 2011] can be made easily decodable by fine-tuning with the class label $C$. We carefully construct our unstructured datasets such that we can take advantage of the theoretical results. Once we can employ $I(X;Y) = H(C)$, the calculation of $H(C)$ can be made trivial by choosing uniformly distributed class labels. Overall, same-class sampling makes it possible to access the true MI values even for unstructured datasets. For convenience, the theorems and proofs are provided in Supplementary B.1.

### 4.4 Generating datasets with adjustable true MI values

By utilizing Theorem 3.1 and 3.2 in Lee et al. [2023], it becomes possible to access the true MI of an unstructured dataset by drawing the positive pairs from a joint distribution $p(x, y)$ where only the class information $C$ is shared by $X$ and $Y$. We first consider a binary random variable $C$ with $p(0) = p(1) = 0.5$. We can design a simple stochastic function that maps $C$ to $X$, where $X$ is an image or sentence embedding. In our benchmark suite, to make use of the error-free classification function, we choose a dataset that easily achieves perfect classification accuracy with a simple classifier (e.g., 1-layer MLP). We adopt the MNIST dataset [Deng, 2012] for $\mathcal{D}_{\text{vision}}$ and BERT [Devlin et al., 2018] fine-tuned sentence embeddings of the IMDB dataset [Maas et al., 2011] for $\mathcal{D}_{\text{NLP}}$. In our implementation, $x$ becomes a sample from $\mathcal{D}$ of class 0 when $c = 0$, and a sample from $\mathcal{D}$ of class 1 when $c = 1$. We design a mapping function from $C$ to $Y$ where a different image or text of same class is drawn. For this basic construction, it can be shown that $I(X;Y) = H(C) = 1$ bit. An image example is shown in Figure 2a. As a concurrent study, Gowri et al. [2024] also utilized the benchmark of 1-bit datasets with MNIST images and protein sequences to evaluate MI estimators.

To construct a dataset with larger MI, two straightforward approaches can be used. In Figure 2b, we combine four samples of Figure 2a to create an image that is four times larger, which means $I(X;Y) = 4$. In Figure 2c, we stack three pairs of samples from Figure 2a and map them to RGB; hence, $I(X;Y) = 3$. We can adopt flexibly use other stratagems to generate a dataset that has a specific value of true MI. Similarly, we generate the text dataset by concatenating the embedding vectors in 1D.

For images, we can insert random samples from other datasets as nuisance to $X$ and $Y$ to make the dataset more realistic without affecting the true MI value, as shown in Figure 2d. Because the

source images remain on top without any occlusion, and there is no fixed relationship between the background chosen for $X$ and the background chosen for $Y$, the nuisance $Z$ does not affect the true $I(X;Y)$. Although strong nuisances might make some samples difficult to recognize the information sources $C$, the true MI is based on the overall distribution, not a few outlier samples. Therefore, the presence of nuisances does not affect the true MI. Further discussion will be provided in Section 5.6. While it is possible to introduce nuisance to any dataset based on the formal definition in Definition 4.3, we applied nuisance only to image datasets in this study, as text datasets already follow practical natural language distributions.

For $\mathcal{D}_{\text{vision}}$ and $\mathcal{D}_{\text{NLP}}$, it is trivial to identify the number of information sources $d_s$ and the representation dimension $d_r$. The number of information sources $d_s$ is always equal to $I(X;Y)$. For instance, Figure 2b has $d_s = 4$. The representation dimension $d_r$ is a design parameter. As the default, we set $d_r = 64^2$ for $\mathcal{D}_{\text{vision}}$ and $d_r = 768 \times 10$ for $\mathcal{D}_{\text{NLP}}$.

### 4.5 Manipulating MI to non-integer values: binary symmetric channel

To manipulate the true MI and construct a dataset with a non-integer MI value, we adopt the concept of binary symmetric channel (BSC) [Cover, 1999]. BSC is a simple and well known form of noisy communication channel in information theory, and we utilize it for scaling down the true MI value in a fully controlled manner. With BSC, $X$ is always consistent with the class variable $C$ but $Y$ is noisy where it is different from $C$ with a crossover probability of $\beta$. Then, the true MI value can be controlled by adjusting $\beta$ between 0 and 0.5.

**Theorem 4.4** (Manipulating MI to be non-integer). *When the information source $C$ is transmitted perfectly to $X$, while it is transmitted to $Y$ over a binary symmetric channel (BSC) with a crossover probability $\beta \in [0, 0.5]$, the mutual information $I(X;Y)$ between $X$ and $Y$ is determined as follows.*

$$I(X;Y) = H(C) \times (1 - H(\beta)) \tag{1}$$

$H(\beta)$ refers to the entropy of a binary variable with the crossover probability $\beta$ [Cover, 1999], and is given by $H(\beta) = -\beta \log \beta - (1 - \beta) \log (1 - \beta)$. When $\beta = 0$, there is no information loss during transmission. Thus, $H(\beta) = 0$ and $I(X;Y) = H(C)$. When $\beta = 0.5$, the channel is completely noisy, and $X$ and $Y$ do not share any information. Thus, $H(\beta) = 1$ and $I(X;Y) = 0$. Any MI value in between can be implemented by choosing an appropriate $\beta$. The proof is provided in Supplementary B.2.

## 5 Empirical investigations

In this section, we investigate seven key aspects that can affect the performance of MI estimators. All investigations are based on our benchmark suite and the empirical findings are reported together. As evaluation metrics, we calculate bias, variance, mean squared error (MSE), and the estimated MI (defined as the average of the estimations) during the training of the critic function. All experiments were conducted on a single NVIDIA GeForce RTX 3090. Detailed experimental setups and raw results are available in Supplementary C and D, respectively.

### 5.1 Choice of critic architecture: superiority of joint critic for unstructured datasets

Poole et al. [2019] observed that using a joint critic outperforms a separable critic for NWJ and JS estimators, while the InfoNCE estimator demonstrating robustness to the choice of critic architecture. For SMILE [Song and Ermon, 2019] estimator, a joint critic surpassed a separable critic in basic Gaussian setups, but this trend reversed in more complex setups of the same dataset. This section aims to extend these insights into the vision and NLP domains, providing guidance on selecting critic architectures across diverse data contexts.

Figure 3 and Figure 9 in Supplementary D present the results of our experiments, which include scenarios where variables share the same domain (image and image, text and text) as well as cases involving cross-domain pairs (image and text). Our key observations are: (1) The joint critic consistently provides reliable estimations across all estimators and data domains; (2) The bilinear critic, while providing stable yet biased estimations for Gaussian datasets, is notably inaccurate in unstructured datasets; (3) The separable critic performs well on unstructured datasets while it often exhibits large variance for Gaussian cases; (4) Contrary to the theoretical proofs and empirical findings of Song and Ermon [2019] on the high variance of the DV estimator in Gaussian datasets, we observe stable performance in unstructured datasets even with large MI values. Notably, no significant

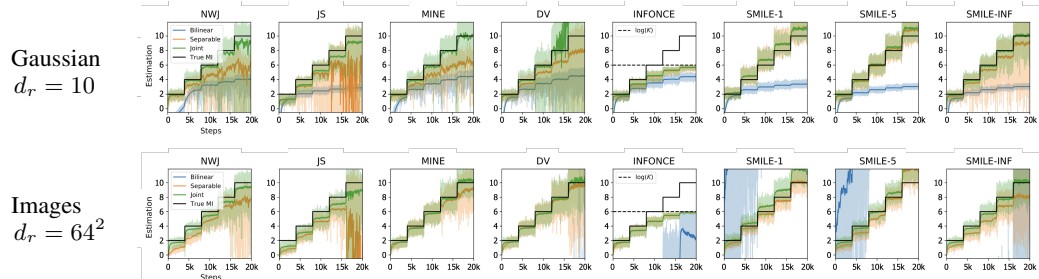

Figure 3: Estimation results for four different benchmarks with $d_s = 10$. Following the experimental setup of Poole et al. [2019], we change the true MI values stepwise and the other hyperparameters are fixed.

advantage of SMILE over DV (or MINE) was observed for both images and sentence embeddings. These findings underscore the pronounced differences in analyzing MI estimators between Gaussian and unstructured datasets, highlighting the robustness of the joint critic in diverse contexts.

## 5.2 Choice of critic capacity: larger capacity does not ensure a higher estimation accuracy

While joint critics often yield the best estimation accuracy in various scenarios, this subsection delves into whether increasing critic capacity could further enhance estimation accuracy. A prior study [Tschannen et al., 2019] posited that larger critic capacities should correlate with more precise estimations. To assess critic capacity, we manipulated the depth

Table 2: Estimation bias/variance/MSE when the true MI is 2 bits. All estimators showed the same trends, so we report the result of the case of SMILE-inf with the joint critic.

| Dataset | Gaussian | | | Images | | | Sentence Embeddings | | |
|---|---|---|---|---|---|---|---|---|---|
| Critic depth | Bias | Variance | MSE | Bias | Variance | MSE | Bias | Variance | MSE |
| 1 | 0.136 | 0.106 | 0.125 | 0.293 | 0.108 | 0.194 | 0.300 | 0.083 | 0.173 |
| 2 | 0.145 | 0.096 | 0.117 | 0.297 | 0.101 | 0.189 | 0.273 | 0.078 | 0.153 |
| 3 | 0.142 | 0.094 | 0.114 | 0.302 | 0.102 | 0.194 | 0.269 | 0.077 | 0.150 |
| 4 | 0.140 | 0.095 | 0.114 | 0.302 | 0.103 | 0.195 | 0.270 | 0.078 | 0.151 |
| 5 | 0.145 | 0.096 | 0.117 | 0.311 | 0.105 | 0.202 | 0.278 | 0.081 | 0.158 |

of the critic network, with specific results for a true MI of 2 bits outlined in Table 2. (Full results are available in Supplementary D.3.)

Contrary to the assertion of Tschannen et al. [2019], our findings reveal an unexpected trend: no discernible positive correlation between critic capacity and estimation accuracy across any data domain. Specifically, the Pearson's correlation coefficient $\rho$ between critic depth and estimation accuracy was $-0.007$ for $\mathcal{D}_{\text{Gaussian}}$, $0.059$ for $\mathcal{D}_{\text{vision}}$, and $-0.001$ for $\mathcal{D}_{\text{NLP}}$. These findings suggest that an increase in critic capacity does not inherently improve estimation accuracy and may even be counterproductive, contradicting previous assumptions and underscoring the need for a nuanced approach to enhancing critic architectures.

Based on our findings, we fixed the critic architecture as the joint critic of 2-layer MLP for all subsequent sections of this study.

## 5.3 Choice of MI estimator: no universal winner exists across the three data domains

Recent advancements have introduced more accurate MI estimators, with notable efforts highlighted in Poole et al. [2019], Song and Ermon [2019], McAllester and Stratos [2020], Cheng et al. [2020]. Among these, the SMILE estimator has been acclaimed for efficiently reducing the estimation variance compared to other estimators, offering a more favorable bias-variance trade-off [Song and Ermon, 2019]. As evidenced in Table 3, the SMILE estimator exhibits a slight superiority over

Table 3: Estimation error (MSE) with the joint critic. For each dataset and true MI, the best cases are marked in **bold**.

| Dataset | True MI (bits) | NWJ | DV | InfoNCE | MINE | SMILE-1 | SMILE-5 | SMILE-inf |
|---|---|---|---|---|---|---|---|---|
| Gaussian | 2 | 0.142 | 0.117 | 0.121 | 0.116 | 0.139 | **0.115** | 0.117 |
| Gaussian | 4 | 0.214 | 0.225 | 0.418 | 0.212 | 0.423 | **0.160** | 0.174 |
| Gaussian | 6 | 0.557 | 1.451 | 2.090 | 0.452 | 0.789 | **0.258** | 0.324 |
| Gaussian | 8 | 4.464 | 456.420 | 7.235 | 0.897 | 1.087 | **0.596** | **0.596** |
| Gaussian | 10 | 8.889 | 1e+07 | 18.400 | 1.973 | 1.443 | 1.658 | **1.262** |
| Images | 2 | 0.288 | 0.175 | 0.179 | 0.217 | **0.142** | 0.191 | 0.189 |
| Images | 4 | 0.357 | 0.233 | 0.479 | 0.250 | 0.338 | **0.229** | 0.239 |
| Images | 6 | 0.577 | 0.366 | 1.912 | 0.340 | 0.854 | **0.210** | 0.372 |
| Images | 8 | 1.058 | 0.787 | 6.457 | **0.602** | 1.278 | 0.659 | 0.694 |
| Images | 10 | **1.580** | 9.529 | 16.742 | 3.249 | 4.197 | 8.987 | 4.899 |
| Text | 2 | 0.150 | 0.143 | 0.162 | 0.147 | **0.097** | 0.155 | 0.153 |
| Text | 4 | 0.336 | 0.312 | 0.641 | 0.335 | **0.234** | 0.301 | 0.328 |
| Text | 6 | 0.668 | 0.596 | 2.390 | 0.625 | **0.305** | 0.328 | 0.606 |
| Text | 8 | 1.309 | 1.092 | 7.288 | 1.108 | 0.297 | **0.274** | 1.159 |
| Text | 10 | 2.659 | 2.345 | 17.757 | 2.027 | **0.319** | 0.749 | 2.476 |

other estimators in Gaussian scenarios and a more pronounced advantage in NLP cases. However,

in vision cases with large true MI values, the NWJ and MINE estimators demonstrate superior performance compared to SMILE. These findings indicate the absence of a universally optimal estimator, highlighting the necessity of context-specific selection for MI estimation across various data domains.

## 5.4 Number of information sources ($d_s$): unstructured datasets outperform Gaussian in handling larger $d_s$

In this subsection, we explore the influence of the number of information sources ($d_s$), as previously defined in Section 4.2, on the accuracy of MI estimation. We incrementally increase $d_s$ from 1 to 100, observing the effects on estimation accuracy. As shown in Figure 4, estimation accuracy deteriorates when $d_s$ becomes excessively large across all data domains.

Interestingly, we observed domain-specific thresholds where MI estimators begin to falter: estimations become unreliable when $d_s$ exceeds 4 in the Gaussian case, 36 in the vision case, and 64 in the NLP case, approximately. This suggests that unstructured datasets, unlike the Gaussian datasets, allow for relatively accurate MI estimations with moderate $d_s$ values within the range of $[4, 36]$.

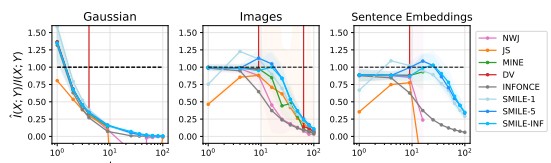

Figure 4: Estimation results varying $d_s$. Shades correspond to the standard deviation of the estimations.

Given that a uniformly distributed classification problem typically involves classes much less than 10M (significantly lower than $2^{36}$), these findings indicate that large $d_s$ values might not be a limiting factor in practical applications.

## 5.5 Representation dimension ($d_r$): it does not affect the estimation accuracy

Real-world datasets can have any representation dimension while having a fixed number of information sources. For example, the image datasets in Figure 2 can be represented in any reasonable dimension without compromising the semantic information. To analyze the MI estima-

Table 4: Estimation results varying $d_r$ for images with $d_s = 1$ and $I(X;Y) = 1$.

| $d_r$ | NWJ | JS | MINE | DV | InfoNCE | SMILE-1 | SMILE-5 | SMILE-inf |
|---|---|---|---|---|---|---|---|---|
| 100 | 0.993 | 0.464 | 0.996 | 0.996 | 0.984 | 0.753 | 0.990 | 0.995 |
| 400 | 0.993 | 0.465 | 0.997 | 0.997 | 0.985 | 0.754 | 0.992 | 0.996 |
| 2500 | 0.994 | 0.465 | 0.997 | 0.997 | 0.986 | 0.753 | 0.992 | 0.997 |
| 10000 | 0.994 | 0.464 | 0.997 | 0.997 | 0.985 | 0.753 | 0.991 | 0.996 |

tion accuracy in these scenarios, we investigate a range of representation dimensions $d_r$ for a fixed number of information sources $d_s$ and the MI value $I(X;Y)$. Specifically, we simply resize the images of size $64^2$ in Figure 2 using a linear interpolation function to obtain images whose size ranges between $10^2$ and $100^2$ while keeping $d_s$ and $I(X;Y)$ fixed.

As shown in Table 4, we observe that the representation dimension does not affect estimation accuracy for images. Even as $d_r$ increases to 10000, we found that almost all the estimators provide accurate estimations. From these results, we conclude that the representation dimension alone does not influence MI estimation accuracy.

## 5.6 Nuisance: MINE turns out to be relatively robust

Nuisance, integral to real-world datasets as defined in Section 4.2, presents unique challenges in MI estimation. To quantitatively assess their influence on images, we place the digits in $x$ over the scaled background image $z \cdot \eta$ as depicted in Figure 5a. We varied the nuisance strength parameter $\eta$ from 0 to 1. Note that introducing nuisance does not alter the true MI values, as class labels remain perfectly predictable when a large number of samples are given. This is the first attempt to investigate how the nuisance affects the estimation of MI.

Our first analysis focused on a fixed true MI of 2 bits, with results detailed in Figure 5b. It was observed that estimation accuracy declines significantly with increased nuisance strength, particularly beyond a threshold of 0.4, across all estimators. Further investigation into larger MI values (Figure 5c) reveals that while large $\eta$ adversely affects estimations for small MI values, MINE and SMILE-inf estimators provide reliable results under moderate nuisance (up to 0.7) and for larger MI values (over

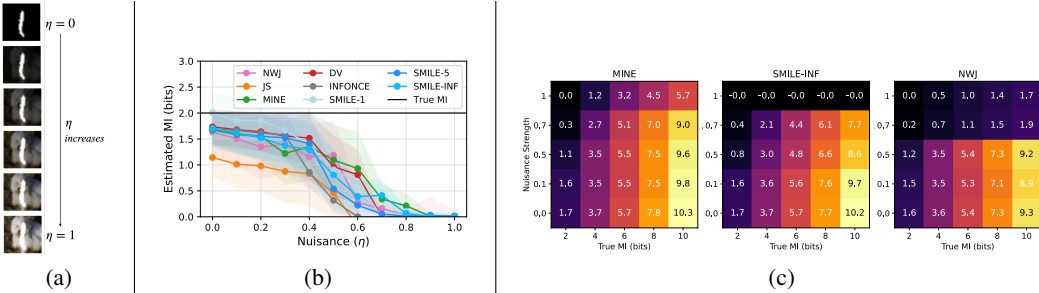

(a)              (b)                       (c)

Figure 5: (a) Example of inserting nuisance to $\mathcal{D}_{\text{vision}}$. (b) Estimation results when the true MI is 2 bits. (c) Estimation results with various values of nuisance strength and true MI for three best-performing estimators. True MI values are on the x-axis and the nuisance strength is on the y-axis.

6 bits). These tight estimations empirically support our theoretical background that $I(X;Y) = H(C)$ when the nuisance exists, as suggested by Theorem 3.1 in Lee et al. [2023]. MINE uniquely offers nonzero estimations even in the presence of the largest nuisance ($\eta = 1$). High nuisance strength may complicate the learning process resulting in increased bias and variance. Consequently, all estimators fail to maintain accuracy under high nuisance conditions, highlighting the need for more robust methods to handle such scenarios.

### 5.7 Network and layer dependency: estimation holds for invertible networks and upper layers

Our final investigation focuses on the accuracy of MI estimators in the context of deep representations (i.e., $I(g(X); g(Y))$, where $g$ represents a deep network) because of the prevalent interest in understanding dependencies between representations rather than raw inputs. While Czyż et al. [2023] highlighted concerns about the reliability of estimators in datasets with tractable distribution functions and under specific unrealistic transformations, our study expands the horizon by examining three invertible networks: MAF [Papamakarios et al., 2017], RealNVP [Dinh et al., 2016], and i-RevNet [Jacobsen et al., 2018] for images and texts. For $\mathcal{D}_{\text{vision}}$, we additionally investigate a non-invertible ResNet-50 network, a widely used non-invertible network, which is pre-trained on the MNIST dataset, to provide more relevant insights for practitioners. This allows us to better align with practical interests in MI estimation beyond invertible networks. Remarkably, as demonstrated in Figure 11 in supplementary material, we found that estimation robustly persists for the representations of $\mathcal{D}_{\text{vision}}$ and $\mathcal{D}_{\text{NLP}}$ across various network architectures.

If deep representations are robust for estimating MI, should this hold across all layers? To address this question, we estimated MI for intermediate layers of ResNet-50 trained on $\mathcal{D}_{\text{vision}}$ without nuisance. Results are summarized in Figure 6. According to the data processing inequality, lower-layer MI cannot be smaller than upper-layer MI. However, estimated MI values indicate the opposite. This discrepancy suggests that MI estimations at lower layers are less precise, whereas upper-layer representa-

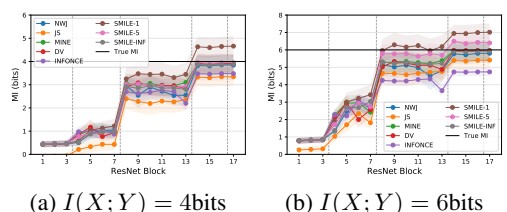

(a) $I(X;Y) = 4$bits    (b) $I(X;Y) = 6$bits

Figure 6: Estimation results for hidden layers of ResNet-50. Dashed lines indicate boundaries between stages. For full results, see Supplementary D.4.

tions yield more accurate estimations. Interestingly, we observe the step-wise estimation results; the transition clearly occurs when the output size changes across all types of estimators. It appears that upper layers might capture abstract, high-level features, potentially offering more meaningful information for MI estimation. In contrast, lower layers might contain more noise and less discriminative features, which could lead to poorer accuracy.

## 6 Discussion and Conclusion

Quantifying complex dependency between variables is an essential topic in machine learning. In this realm, sample-based neural MI estimators have been the primary choice for many deep learning

applications. The MI estimators have been directly used for improving downstream task performance or indirectly used for motivating learning method developments. However, there has been hardly any attempt to evaluate the accuracy of these MI estimators over real-world datasets such as images and texts. In this study, we proposed a novel benchmark suite for evaluating neural MI estimators on unstructured datasets, where the underlying distribution functions are not accessible. Our findings reveal discrepancies in estimation accuracy between traditional Gaussian benchmarks and unstructured data scenarios, highlighting the limitations of Gaussian benchmarks in capturing the nuances of MI estimation in practical settings. Notably, our findings on unstructured datasets demonstrate that MI estimators can yield remarkably accurate results, particularly in conjunction with deep representations, indicating their potential to continue driving advancements in deep learning research. While our study does not cover the entire spectrum of real-world datasets, it signifies a substantial step forward in evaluating and understanding MI estimators beyond purely statistical datasets. We hope that this benchmark suite not only offers a new standard for evaluating MI estimators but also catalyzes further research, enriching our comprehension of MI across a diverse data domains.

**Acknowledgement**   This work was supported by the following grants funded by the Korea government: NRF (NRF-2020R1A2C2007139, NRF-2022R1A6A1A03063039) and IITP ([NO.2021-0-01343, Artificial Intelligence Graduate School Program (Seoul National University)], [No. RS-2023-00235293]).

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

## A  Broader Societal Impact Statement

This paper introduces a new benchmark suite for evaluating mutual information (MI) estimators, particularly in unstructured datasets like images and texts. The broader impacts of our work are significant in two folds. First, by providing a more realistic and challenging benchmark for MI estimators, this research can lead to the development of more accurate and robust estimation methods, especially in complex data scenarios. Second, improved MI estimation methods can benefit fields such as computer vision and natural language processing, where understanding intricate data relationships is crucial, thereby enhancing the efficiency and effectiveness of AI systems in practical applications. In essence, our research has the potential to influence various aspects of society through the improved understanding and application of mutual information in complex data domains.

## B  Theorems

### B.1  Theorems and proofs

In Section 4.3, we adopt the same-class sampling method for positive pairing suggested in [Lee et al., 2023]. For convenience, we provide the theorems and proofs in [Lee et al., 2023] as follows.

**Proposition B.1** (InfoNCE estimation as a lower bound of the true MI [Oord et al., 2018, Poole et al., 2019]). *The InfoNCE estimation of mutual information is a lower bound of the true mutual information.*

$$\hat{I}(X;Y) \leq I(X;Y) \tag{2}$$

**Proposition B.2** ($\log(2K-1)$ Bound [Oord et al., 2018, Poole et al., 2019]). *The InfoNCE estimation of mutual information is upper bounded by $log(2K-1)$.*

$$\hat{I}(X;Y) \leq \log(2K-1) \tag{3}$$

*Proof.* The proof is based on the variational bound derivation. Let $q(x|y) = \frac{p(x)}{Z(y)}e^{z_{x,y}/\tau}$, where $Z(y) = \mathbb{E}_{p(x)}[e^{z_{x,y}/\tau}]$. Then the true MI, $I(X;Y)$, can be bounded as the following.

$$I(X;Y)$$

$$= \mathbb{E}_{p(x,y)}\left[\log\frac{p(x,y)}{p(x)p(y)}\right] = \mathbb{E}_{p(x,y)}\left[\log\frac{p(x|y)}{p(x)}\right] \tag{4}$$

$$= \mathbb{E}_{p(x,y)}\left[\log\frac{q(x|y)}{p(x)}\right] + \mathbb{E}_{p(y)}[KL(p(x|y)||q(x|y))] \tag{5}$$

$$\geq \mathbb{E}_{p(x,y)}\left[\log\frac{q(x|y)}{p(x)}\right] \tag{6}$$

$$= \mathbb{E}_{p(x,y)}\left[\log\frac{e^{z_{x,y}/\tau}}{Z(y)}\right] \tag{7}$$

$$\approx \mathbb{E}\left[\log\frac{e^{z_{x_i,y_i}/\tau}}{\frac{1}{2K-1}\sum_{j=1}^{K}\left(\mathbb{1}_{[j\neq i]}e^{z_{x_i,x_j}/\tau} + e^{z_{x_i,y_j}/\tau}\right)}\right] \tag{8}$$

$$= \log(2K-1) \tag{9}$$

$$+ \mathbb{E}\left[\log\frac{e^{z_{x_i,y_i}/\tau}}{\sum_{j=1}^{K}\left(\mathbb{1}_{[j\neq i]}e^{z_{x_i,x_j}/\tau} + e^{z_{x_i,y_j}/\tau}\right)}\right] \tag{10}$$

$$= \log(2K-1) - \mathcal{L}_{\textit{InfoNCE}} \tag{11}$$

$$\triangleq \hat{I}(X;Y) \tag{12}$$

The inequality in Eq. (6) is due to the non-negativeness of KL-divergence, and the approximation in Eq. (8) is due to the replacement of the expectation with its empirical mean.

The proof of Proposition B.1 is directly obtained from the above derivation of InfoNCE variational bound. The proof of Proposition B.2 also follows from the derivation. In Eq. (11), the term $-\mathcal{L}_{InfoNCE}$ is always negative because the argument of the $\log$ term in Eq. (10) is always between zero and one. This can be easily confirmed because the denominator term $\sum_{j=1}^{K} \left( \mathbb{1}_{[j \neq i]} e^{z_{x_i, x_j}/\tau} + e^{z_{x_i, y_j}/\tau} \right)$ is a sum of positive values and because the summation includes the numerator term $e^{z_{x_i, x_j}/\tau}$. $\square$

**Proposition B.3.** *For the same-class sampling $\mathcal{T}_{class}$ with its joint distribution $p_{class}(x, y)$, the mutual information between the first view $X$ and the second view $Y$ is **upper bounded** by the class entropy.*

$$I_{class}(X; Y) \leq H(C) \tag{13}$$

*Proof.* From the construction of same-class sampling, the dependency can be expressed as $X \leftarrow C \rightarrow Y$. The dependency is Markov equivalent to $X \rightarrow C \rightarrow Y$ because both Markov chains encode the same set of conditional independencies. Then,

$$I(X; Y) \leq I(X; C) \tag{14}$$
$$= H(C) - H(C|X) \tag{15}$$
$$\leq H(C), \tag{16}$$

where the first inequality follows from the data processing inequality [Cover, 1999] and the second inequality follows from the entropy's positiveness for discrete random variables. $\square$

With Proposition B.1 and Proposition B.3, the following main theorem can be obtained.

**Theorem B.4.** *For the same-class sampling $\mathcal{T}_{class}$ with its joint distribution $p_{class}(x, y)$, the mutual information between the first view $X$ and the second view $Y$ is **equal to** the class entropy when the estimated mutual information is equal to the class entropy.*

$$\hat{I}_{class}(X; Y) = H(C) \tag{17}$$
$$\Rightarrow I_{class}(X; Y) = H(C) \tag{18}$$

*Proof.* The following is a direct result of Proposition B.1 and Proposition B.3.

$$\hat{I}_{class}(X; Y) \leq I_{class}(X; Y) \leq H(C) \tag{19}$$

Because the true MI $I_{class}(X; Y)$ is in the middle, $\hat{I}_{class}(X; Y) = H(C)$ means that all three are of the same value. $\square$

For the most popular downstream benchmarks, such as CIFAR and ImageNet, the calculation of $H(C)$ is trivial. Now, thanks to the Theorem B.4, we can identify the true MI with no ambiguity whenever $\hat{I}_{class}(X; Y) = H(C)$ is satisfied. This theorem will be utilized as a key enabler for a rigorous MI analysis.

Theorem B.4 can be useful when the true MI value is required for an MI analysis. The theorem, however, is not useful when the condition $\hat{I}_{class}(X; Y) = H(C)$ is not satisfied. For such a case, we derive another equality that can be proven under an error-free classifier assumption.

**Theorem B.5.** *For the same-class sampling $\mathcal{T}_{class}$ with its joint distribution $p_{class}(x, y)$, the mutual information between the first view $X$ and the second view $Y$ is **equal to** the class entropy when there exists an error-free classification function $f_{class}(\cdot)$.*

$$\exists \text{ An error-free classifier } f_{class}(\cdot) : X \rightarrow C \tag{20}$$
$$\Rightarrow I_{class}(X; Y) = H(C) \tag{21}$$

In reality, such an error-free classification function $f_{class}$ does not exist for practical and interesting problems. Nonetheless, the equality result is useful for understanding MI in a high-accuracy regime. The proof is provided below.

*Proof.* From the construction of same-class sampling, the dependency can be expressed as $S \to C \to X$ and $S \to C \to Y$ where $C$ is the class label of the sampled source image $S$. Because of the error-free classifier $f_{\text{class}}(\cdot)$, the class label information can be perfectly extracted from $X$ or $Y$. This means that $X \to C$ and $Y \to C$ also hold. Using the dependencies, we can conclude that the following is a valid Markov chain.

$$S \to C \to X \to C \to Y \to C \tag{22}$$

The desired equality proof can be obtained by deriving an upper bound $I(X;Y) \leq H(C)$ and a lower bound $H(C) \leq I(X;Y)$. The upper bound follows directly from the Proposition B.3. The lower bound can be derived by applying the data processing inequality to the Markov dependency $C \to X \to Y \to C$ that can be confirmed from Eq. (22).

$$I(C;C) \leq I(X;Y) \tag{23}$$
$$\Rightarrow H(C) \leq I(X;Y) \tag{24}$$

Note that $I(C;C)$ is the self-information that is equal to $H(C)$. $\qquad\square$

### B.2 Proof of Theorem 4.4: Detailed explanation for binary symmetric channel (BSC)

Figure 7: Construction of an image dataset with a non-integer MI value. We utilize binary symmetric channel to corrupt the label of $Y$.

*Proof.* In Section 4.5, we introduced binary symmetric channel (BSC) [Cover, 1999] to construct a dataset with a non-integer MI value. We first consider the basic construction case of Figure 2a where $H(C) = 1$. As shown in Figure 7, the transmission process of BSC for $C \to Y$ corresponds to a binary channel where the the label of $Y$ is corrupted with probability $\beta$. Then, the mutual information can be evaluated as follows.

$$\begin{aligned}
I(X;Y) = I(C;Y) &= H(C) - H(C|Y) \\
&= 1 - \sum p(y) H(C|Y = y) \\
&= 1 - \sum p(y) H(\beta) \\
&= 1 - H(\beta)
\end{aligned}$$

The first equality comes from the Markov equivalence between $X$ and $C$. Note that $H(\beta)$ is symmetric and it can be expressed as $H(\beta) = -\beta \log \beta - (1 - \beta) \log (1 - \beta) = H(1 - \beta)$. For the case where $H(C)$ is an integer larger than 1, we can apply the above BSC trick to each binary information source. Then, we obtain the general result of $I(X;Y) = H(C) \times (1 - H(\beta))$. For instance, when we have three independent binary information sources, $H(C) = 3$ and $I(X;Y) = 3 \times (1 - H(\beta))$ by applying BSC with the same parameter $\beta$ to each binary information source. $\qquad\square$

## C Detailed experimental setups

We follow the setup of the case of multivariate Gaussian [Belghazi et al., 2018, Tschannen et al., 2019, Song and Ermon, 2019, Poole et al., 2019]. For Gaussian datasets, the critic network $f(x, y)$ is trained to maximize $\hat{I}(X;Y)$ with the real-time-generated inputs $X$ and $Y$. For images and texts, the critic network $f(x, y)$ is trained to maximize $\hat{I}(X;Y)$ using a limited number of inputs $X$ and $Y$, specifically 50,000 samples. The positive pairs, *i.e.,* samples from the joint distribution $p(x, y)$, are randomly sampled from the limited number of inputs, thus ensuring that the diversity of the paired samples is much larger than the size of inputs. When the other factors are fixed, only the true MI

$(I(X;Y))$ is controlled during estimation. A complete estimation occurs over 20k steps, and we vary the true MI ($I(X;Y)$) over time. For calculating the estimated values, bias, variance, and MSE, we use all the estimations during 4000 steps with same true MI values.

Note that our method is not restricted to cases where $X$ and $Y$ have the same dimensionality. When $X$ and $Y$ have different dimensions (*e.g.*, a 4096-dimensional image and a 7680-dimensional sentence embedding as in Figure 9(Mixture)), we handle this by expanding the smaller dimension with redundant information, essentially copying parts of the original vector. This ensures no information loss and allows both $X$ and $Y$ to contribute equally to the training of the critic functions. For instance, in the case mentioned, the first 3584 dimensions of the 4096-dimensional image can be copied to expand X to 7680 dimensions, aligning it with Y.

We provide the architecture details for the critic network as follows. We reference the official code of [Tschannen et al., 2019, Song and Ermon, 2019]. Critic functions model the relationship between all pairs of $(x_i, y_j) \ \forall i, j \in [1, K]$. Variational bounds approximate MI by using the diagonal terms as the values from the joint distribution $p(x, y)$ and the off-diagonal terms as the values from the marginal distribution $p(x)p(y)$. For the separable critic $f(x_i, y_j) = f_1(x_i)^T f_2(y_j)$, we use the same architecture for $f_1$ and $f_2$ as a 2-layer MLP with 256 units and 32-dimensional outputs. For the concatenated critic $f(x_i, y_j) = f([x_i, y_j])$, we use 2-layer MLP with 256 units. To train the critic networks, we set the batch size $K$ as 64. We optimize the variational bounds of mutual information using Adam [Kingma and Ba, 2014] with a learning rate of 0.0005. The detailed codebase and datasets are available in `https://github.com/kyungeun-lee/mibenchmark`.

# D    Additional experimental results

All the raw results, in addition to the codebase and materials are available in `https://github.com/kyungeun-lee/mibenchmark`.

## D.1    Changing embedding networks for NLP dataset

For sentence embeddings, we replaced the BERT encoder with DistilBERT [Sanh et al., 2019] and RoBERTa [Liu et al., 2019] on the same dataset. As shown in Figure 8, estimation holds for the deep representations regardless of which embedding encoder is used. We also found that all the variational MI estimators work well for all cases.

From our findings, we conclude that the variational MI estimators are unexpectedly accurate for images and sentence embeddings datasets, and also for their deep representations.

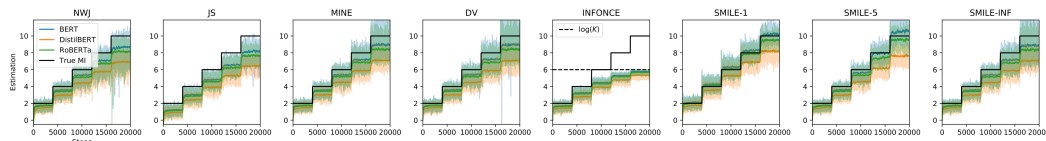

Figure 8: Estimation results for different types of embedding encoders for sentence embeddings. For all types of embedding encoders, we achieve accurate estimations.

## D.2 Additional results of Section 5.1

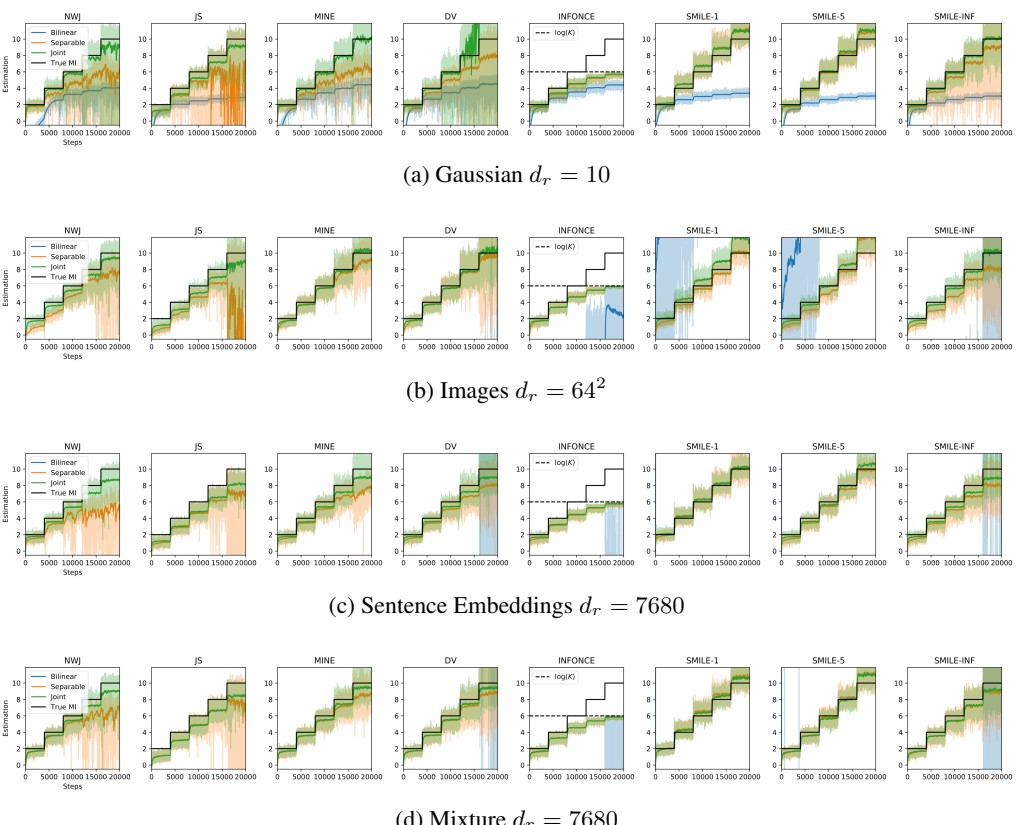

(a) Gaussian $d_r = 10$

(b) Images $d_r = 64^2$

(c) Sentence Embeddings $d_r = 7680$

(d) Mixture $d_r = 7680$

Figure 9: Estimation results for four different benchmarks with $d_s = 10$. Following the experimental setup of Poole et al. [2019], we change the true MI values stepwise and the other hyperparameters are fixed.

## D.3 Additional results of Section 5.2

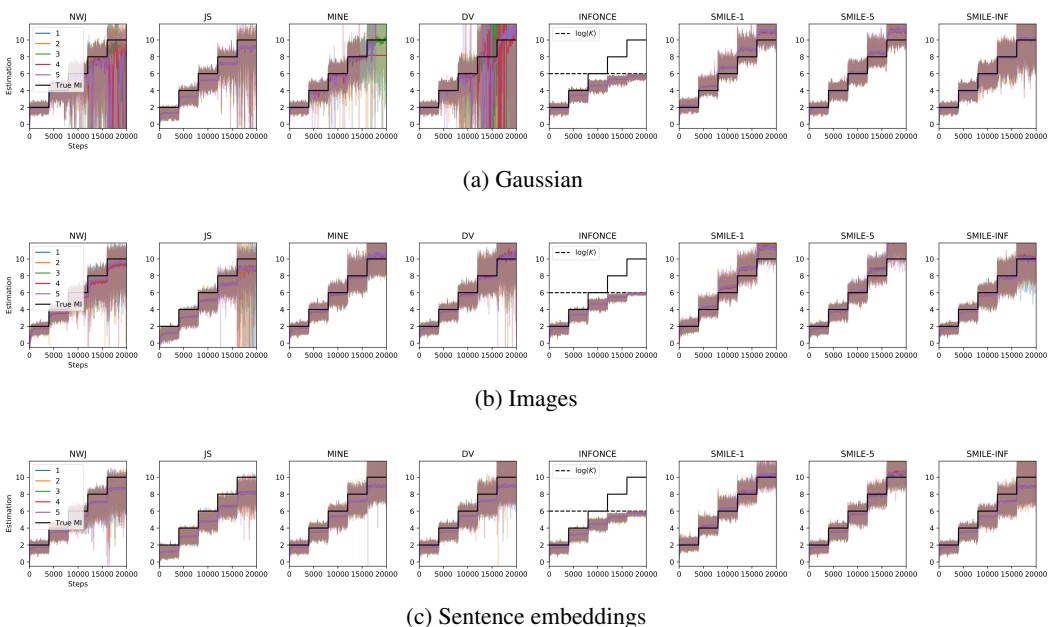

(a) Gaussian

(b) Images

(c) Sentence embeddings

Figure 10: Full results for Table 2. We calculate the bias, variance, and MSE for each true MI values in the manuscript.

## D.4 Additional results of Section 5.7

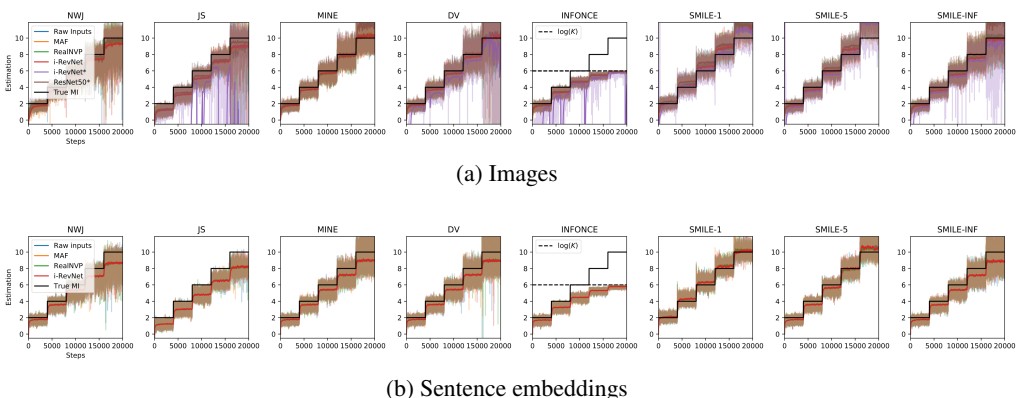

(a) Images

(b) Sentence embeddings

Figure 11: Estimation results for deep representations. For simplicity, we use the deep networks at random initialization except for the ResNet-50, which is pre-trained on the MNIST dataset.

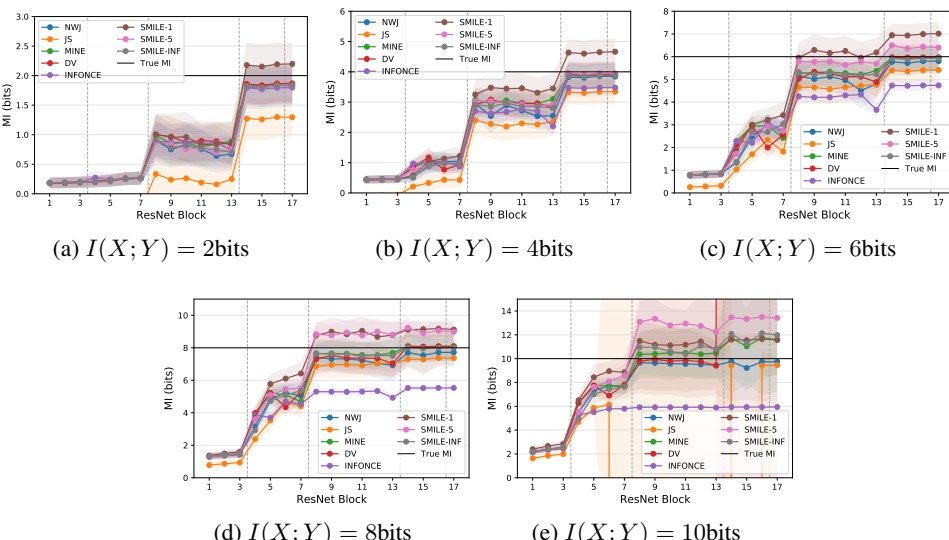

(a) $I(X;Y) = 2$bits     (b) $I(X;Y) = 4$bits     (c) $I(X;Y) = 6$bits

(d) $I(X;Y) = 8$bits     (e) $I(X;Y) = 10$bits

Figure 12: MI estimation results for hidden layers of ResNet-50.

