# OpenReview forum: "A Benchmark Suite for Evaluating Neural Mutual Information Estimators on Unstructured Datasets"
_NeurIPS.cc/2024/Datasets_and_Benchmarks_Track — NeurIPS 2024 Track Datasets and Benchmarks Poster_

### Official Review · Reviewer_GBdp · 2024-07-26

**Rating:** 6
**Confidence:** 4
**Clarity:** Beyond what I have discussed already,…

**Review:**

In general, I consider this article to be well-posed, provide useful insights, as well as of being of interest for the community. Certainly, testing MI estimators in other data than Gaussians (while knowing the ground-truth) is a valuable tool to have, and this work opens the venue for more complex benchmarks to be developed. While I think there is room for improvement (more details below), I am generally positive with this submission.

I would be happy to increase my score if necessary after the rebuttal.

**Strengths:**

- This work opens the venue for testing MI estimators in more realistic benchmarks, which can be really helpful for the community.
- The way of constructing the datasets while having fine control on the MI is quite interesting and can be easily applied to other datasets.
- While the writing could be improved at times, it does a good job at motivating the problem and explaining every taken step.
- The insights provided by the empirical results are interesting and relevant.

**Additional Feedback:**

- Explain better why the non-invertible network in section 5.7.
- In 5.6, it is rather arbitrary why to call some nuisances more "severe" than others, as none of them affect the MI. The authors should clarify what they mean by severe here.
- It should be re-stated how the dimensionality is being increased in section 5.5.
- Please user proper citations (i.e. `\citet`) when citing within the text. (e.g. line 199).
- The notation of equation 1 is quite unclear. What is the entropy of a fixed variable $\beta$? Consider using a different notation or explaining the existing one.

**Correctness:**

While I have only fully read the main manuscript, all the explanations and descriptions are sound (beyond what I have already discussed above), and I don't have any major concerns when it comes to the parts described there.

**Documentation:**

From a quick glimpse to the source code and appendices, I do believe that there are enough details to reproduce the results reported in this work.

**Ethics:**

I do not have concerns in this regard.

**Limitations:**

I think the limitations are well discussed for the most part.

**Opportunities For Improvement:**

- Figures and tables are way too small.
- Some details are underexplained, e.g., it is unclear at first why and how using a Resnet in section 5.7. A similar problem occurs when explaining the MI estimators, e.g., lines 78 and 79 and completely uninformative for someone not familiar with these methods.
- In line 91 it is said that the results reported here contradict that from other work. Have you tried using their approximated metrics to see if the results then agree? I consider this a necessary sanity check.
- I disagree with the entire discussion in section 5.5, increasing the dimensionality does not necessarily increase the sparsity (specially in image data). Actually, I think sparsity is the reason why the estimators on the Gaussian data in section 5.4 work as poorly (the samples of Gaussians in high-dimensions are extremely concentrated on their surface, making them super sparse).
- Similarly, I think the conclusions from Section 5.2 are a bit too strong. As the authors say, training becomes more difficult with the depth, but does not mean that a well-trained model does not improve the estimations. The authors should put emphasis on the training issues, or ensure that the models well properly optimized.

**Relation To Prior Work:**

From what I have read, I do believe prior work is properly discussed (note, however, that this is not my main field of research).

**Summary And Contributions:**

This work proposes a benchmark for evaluation mutual information (MI) estimator on more realistic scenarios. The motivation of the paper is that most research has focused on analytically evaluating MI estimator on Gaussian data (as it has a closed-form solution), yet they have proven to also being useful in practice. To understand the capabilities of these estimators on more realistic data, while having control over the ground-truth MI, this work leverages existing results to construct artificial datasets (based on true ones) so that they can control the true MI of the variables. In this way, the authors evaluate the performance of existing estimators in unstructured data for a number of controllable parameters, providing new insights into what factors determine the effectiveness of existing methods.

---

> ### Author Rebuttal · Authors · 2024-08-16
>
> [Q5] **Section 5.7 - “Explain better why the non-invertible network in section 5.7.”**
> - Thank you for highlighting this point. In the previous works, invertible networks were frequently analyzed because they can be proven to preserve the equality $I(X;Y)=I(g(X);g(Y))$ where $g$ represents a deep network. This allows interesting validations to be performed for deep representations.  In our work, we intentionally expanded the scope to include a non-invertible ResNet-50 network to provide more relevant insights for practitioners. Because ResNet-50 is widely used in practical applications, investigating them in addition to the invertible networks allows us to better align with the practical interest in estimating MI. As shown in Figure 10 of Appendix D.3, we have found similar results between the invertible networks and ResNet-50. To improve clarity, we will revise the manuscript to explicitly state the rationale for using a non-invertible network in Section 5.7.
>
> [Q6] **Section 5.6 - “it is rather arbitrary why to call some nuisances more "severe" than others, as none of them affect the MI.”**
> - Thank you for highlighting this issue. You are correct that since the nuisance does not affect the MI, the term "severe(strong)" may be misleading. We will revise the manuscript to replace "severe(strong)" with a more precise expression, such as "large $\eta$", to accurately convey the intended meaning, in lines 169, 294, and 300.

---

> ### Author Rebuttal · Authors · 2024-08-16
>
> Thank you for your thorough feedback. Our point-by-point answers to the reviewer’s concerns and questions are provided below.
>
> [Q1] **Clarity of writing**
> - Thank you for the feedback. We will enhance the clarity and readability of the manuscript by addressing the following points:
>
>    - Figures and Tables: We will increase the size of figures and tables to ensure they are easily readable. For instance, Figure 3 includes too many plots for a limited space, and the figures and the texts in the figures become too small. To address this issue, we will move the mixture results in the last row to the supplementary material and increase the size of the figures and text.  In Section 5, the font size of tables and the x/y-axis names of the figures will be increased in the revised manuscript.
>
>    - Details and Explanations: The introduction of ResNet in Section 5.7 will be addressed as explained in the response to [Q5] below. As for lines 78-79, the explanations of the variational bounds in lines 78-79 will be moved to the supplementary materials with further details.
>
>    - Dimensionality in Section 5.5: We will re-state and clarify how dimensionality is increased in Section 5.5. Specifically, we will provide an explanation based on Definition 4.2 and the writings in lines 172-175. In Section 5.5, we simply resize the images of size $64^2$ in Figure 2 using a linear interpolation function to obtain images whose size ranges between $10^2$ and $100^2$.
>
>    - Proper Citations (i.e. \citet): We will use proper citations to ensure correct referencing throughout the text, including lines 28, 84, 89, 92, and 199.
>
>    - Notation of equation 1 ($H(\beta)$): $H(\beta)$ refers to the entropy of a binary variable with the crossover probability $\beta$, and this is a common notation in information theory [1]. However, we recognize that not all readers may be familiar with this notation, and we will add a detailed explanation to make this clearer -- $H(\beta)=-\beta\log{\beta}-(1-\beta)\log{(1-\beta)}$. Thanks for your comment.
>
> [Q2] **“In line 91 … I consider this a necessary sanity check.”**
> - Thank you for the comment. We can address this question in two parts:
>    - In lines 91-92, we mentioned that “the analysis [7] can be highly misleading because they evaluated based on the approximated metrics instead of the true MI.” We apologize for the confusion this may have caused, as it may have suggested a contradiction between our results and those of [7]. Our intention was to caution against relying solely on evaluation benchmarks that do not use true MI values and not to criticize the benchmark design itself. The self-consistency tests in [7] are valid, and an accurate estimator should yield ideal outcomes as explained by the authors. We will revise the statement to say, “the approach in [7] is limited because only relative relationships can be examined as opposed to directly comparing against the ground truth MI values”, to avoid further misunderstanding.
>    - Regarding the suggestion of a sanity check against the approach in [7], our method and the self-consistency tests in [7] are both valid. Our method, however, is far superior because we have access to the true MI values. For the common experiments between our work and [7], we can directly compare the results. For example, SMILE estimators demonstrate good accuracy across various findings in Section 5 of our study and in the three types of self-consistency tests in [7]. On the other hand, while MINE performs well on unstructured datasets in our study, it shows poor performance on the self-consistency tests, such as additivity, in [7].
>
> [Q3] **Section 5.5 - “increasing the dimensionality does not necessarily increase the sparsity”**
> - We sincerely apologize for the misleading statement regarding sparsity. You are correct that increasing dimensionality does not necessarily increase sparsity, especially in image data. Our choice of the word “sparsity” was incorrect and misleading. In lines 273-281, we will revise the text to remove the word “sparsity”. The revised text will state: “Real-world datasets can have any representation dimension while having a fixed number of information sources. For example, the image datasets in Figure 2 can be represented in any reasonable dimension without compromising the semantic information. To analyze the MI estimation accuracy in these scenarios, we investigate a range of representation dimensions $d_r$ for a fixed number of information sources $d_s$​ and the MI value$I(X;Y)$. Specifically, …”
>
> [Q4] **Section 5.2 - “Similarly, I think the conclusions from Section 5.2 are a bit too strong”**
> - We sincerely apologize for the misleading statement in lines 234-235, "Larger capacity can lead to unstable learning dynamics, which can increase both bias and variance." As shown in Table 2 and Figure 9, larger capacity does not increase the estimation variance at all. While our conclusion that “a larger capacity does not guarantee a higher estimation accuracy” remains valid, the explanation provided in lines 234-235 was incorrect. So, we will remove the statements in lines 234-235 for a better clarity.

---

> ### Author Response · Authors · 2024-08-29
>
> Dear Reviewer, if you have any additional concerns, we would greatly appreciate it if you could share them before the discussion period ends. Thank you!

---

> > ### Comment · Reviewer_GBdp · 2024-08-31
> >
> > Dear authors, my apologies for the late response. I do appreciate the detailed response, and I truly believe that this work will be significantly improved after applying the changes discussed during the rebuttal period.
> >
> > Therefore, I am updating my score to reflect these changes. Thanks for the effort!
> >
> > PS: I cannot change it anymore on OR, but I trust the AC will see the rebuttal discussion and thus the improvements.

---

> > > ### Author Response · Authors · 2024-09-01
> > >
> > > Dear Reviewer GBdp, thank you for your comments. We will update the manuscript to the best of our abilities.

---

### Official Review · Reviewer_7BAF · 2024-08-06
**Strong paper on evaluating neural MI estimators**

**Rating:** 7
**Confidence:** 3
**Clarity:** The paper was well-written and was ea…

**Review:**

Overall the work is well-motivated by the need to evaluate neural MI estimators in situations beyond multivariate Gaussian data distributions (i.e. unstructured data use cases). The theoretical and empirical investigations are thorough and easy to follow.

**Strengths:**

- The problem is well-motivated and novel in that it allows for a unified suite of tests for evaluating neural MI estimators on unstructured data.
- The authors analyze different factors that affect MI estimators using their benchmarking suite for text and image data.
- The authors provide methods of constructing datasets with different ground truth MI values for the base datasets chosen (i.e., MNIST, IMDB), including non-integer MI values
    - The authors provide good theoretical backing for this and other theorems

**Additional Feedback:**

- Table 1 is not easy to read, especially the rows spanning multiple columns.
- Line 119 says that you focus on only 3 of the 7 factors, but the results include all 7. Are you only providing definitions for 3? This could be slightly clearer.
- For Definition 4.2, what if $X$ and $Y$ are not of the same size?

**Correctness:**

The evaluation methods and experiments seem appropriate. No major issues were noticed.

**Documentation:**

There is sufficient detail for reproducibility and a provided code repository with a quality README.

**Ethics:**

No ethical concerns

**Limitations:**

The only limitation of the work that the authors discussed is that their study doesn't cover the entire spectrum of real-world datasets.

**Opportunities For Improvement:**

This may be a bit out of scope for the work, but it would be interesting to see empirically how the methods compare for a dataset whose class information isn't as easily decodable (e.g. not MNIST). There are a lot of domains where datasets don't have methods for achieving near-perfect performance (unless it is some degenerate or truly trivial dataset) and it would be interesting to know how far the "easily decodable" assumption can be stretched.

**Relation To Prior Work:**

The authors discussed related and prior work and mentioned how this work improved upon it.

**Summary And Contributions:**

This work provides a benchmarking suite for evaluating neural MI estimators for unstructured data (e.g. images, text), which goes beyond the traditional multivariate Gaussian benchmarks used in prior work. The goal is to shed some light on the reliability of what can impact neural MI estimators and be used to benchmark and evaluate new MI estimators.

---

> ### Author Rebuttal · Authors · 2024-08-16
>
> Thank you for your helpful feedback. Our point-by-point answers to the reviewer’s concerns and questions are provided below.
>
> [Q1] **Dataset whose class information isn't as easily decodable**
>    - Thank you for your insightful comment. As addressed in Theorem B.5 of the manuscript, we use the term “easily decodable” to mean that “there exists an error-free classification function $X\rightarrow C$.” Theoretically, this assumption can be satisfied for practical datasets where the label information $C$ is fully accessible. A related discussion is presented in [21] (Section IV, “B. MI stands as a superior measure for representation quality”), where a variety of practical image datasets are analyzed and a positive correlation between classification accuracy and estimated MI is found. If we interpret “easily decodable” in terms of classification accuracy on $C$, indeed there could be interesting relationships between decodability and MI estimation. However, as the reviewer correctly points out, defining what constitutes an "easily decodable" dataset is not straightforward and requires more rigorous and thorough theoretical and empirical exploration. This is a valuable area for future research, but it is beyond the scope of our current study.
>
> [Q2] **Clarity of writing**
>    - As for Table 1, we made the mistake of trying too hard to save space for the rest of the writing. We will update the table for a better readability.  Additionally, we will enhance the caption of Table 1 to improve clarity, for example, by specifying that “DV, NWJ, and InfoNCE utilize the same formulation for optimization and estimation.”
>    - As for line 119, we apologize for the confusion caused by the multiple uses of the term “factor” in the manuscript. To clarify, we will use distinct terms for each context: “factor” will refer to the three essential factors discussed in Section 4.2; “finding” will refer to the seven key findings presented in Section 5; and “hyperparameters” will be used to describe the experimental setups in Figure 3.
>    - Thank you for the feedback.
>
> [Q3] **Definition 4.2, what if  $X$ and $Y$ are not of the same size?**
>    - Thank you for raising this point. Our method is not restricted to cases where X and Y have the same dimensionality. When X and Y have different dimensions (e.g., a 4096-dimensional image and a 7680-dimensional sentence embedding as in Figure 3(Mixture)), we handle this by expanding the smaller dimension with redundant information, essentially copying parts of the original vector. This ensures no information loss and allows both X and Y to contribute equally to the training of the critic functions. For instance, in the case mentioned, the first 3584 dimensions of the 4096-dimensional image can be copied to expand X to 7680 dimensions, aligning it with Y. We will add this description to the supplementary material.

---

> > ### Comment · Reviewer_7BAF · 2024-08-21
> > **Response to rebuttal**
> >
> > Thank you for you clarifications and responses. I will maintain my score.

---

> > > ### Author Response · Authors · 2024-08-22
> > >
> > > Dear reviewer 7BAF, thanks for your note. - Authors

---

### Official Review · Reviewer_AxEX · 2024-08-07
**Review for Submission969**

**Rating:** 7
**Confidence:** 3
**Correctness:** Yes, the claims are correct and suppo…

**Review:**

The paper makes significant contributions towards developing a better understanding of neural MI appraoches. In particular, it provides insight towards the scenarios under which these methods may provide accurate estimates of the true MI. For example, they find that critic capacity does not significantly impact the performance of MI methods, which would may help future researchers test their methods for MI estimation with lower resource requirements. While the mechanism to construct datasets with specific MIs are not themselves original (relying on the authors' previous work [21] and on well-known results from information theory), the MI methods evaluated and the insights provided are original. However, the section introducing the theoretical background critic-based approaches to MI do not have very clear exposition and could be improved.

**Strengths:**

S1: The findings in this paper help provide an easy to construct and inexpensive to evaluate benchmark for new neural MI methods, thus helping future research in this domain.

S2: The methods for constructing datasets with specific MI are well-motivated theoretically with proofs provided.

S3: The work provides some prescriptive results (eg. MINE is relatively robust to noise) that may help future research that utilizes MI methods as a part of their work.

**Additional Feedback:**

F1: To improve the clarity of the paper, the authors should only include representative plots/summary statistics in the main body and move additional results to the supplementary.

F2: For section 5.3, it seems like SMILE-5 is either the best or close to the best method in all cases. Does this result really support the conclusion of no universal winner?

**Clarity:**

C1: The clarity of the background materials and the empirical evidence could be improved.

**Documentation:**

Yes.

**Ethics:**

No ethics concerns.

**Limitations:**

Depending on whether there exists a method to add nuisance noise to text datasets, this could be a limitation of the proposed approaches for benchmark construction.

**Opportunities For Improvement:**

W1: The previous work on expanding MI benchmarks evaluates beyond Gaussian distributions (reference [20] in the paper) evaluates on non-critic approaches as well. It is unclear why these methods are not included in these benchmarks.

W2: While the authors describe a method to add nuisance in image data, there is no such method provided for text datasets.

W3: The clarity of writing could be improved in the background section. Further, a large number of plots are included for some major findings (eg 32 plots in Figure 3) which make it difficult to ascertain whether all the results match the provided findings.

**Relation To Prior Work:**

Yes.

**Summary And Contributions:**

This paper introduces a benchmark to compare neural approaches for measuring mutual information (MI) on text and image datasets. While previous work has extended MI beyond Guassian distributions to other tractable distributions, the authors claim that their work is the first to create a "real-world" benchmark. The authors create these datasets though three mechanisms, pairing data from the same class (so the MI equals the entropy of the class distribution under mild assumptions), adding random background as noise, and through binary symmetric channels. These mechanisms allow the authors to control the number of information sources, the representation dimension, and noisance noise parameters for non-Gaussian settings. The first two mechanisms allow controlling for MI to arbitrary integer values, while the third mechanism allows controlling for MI to equal non-integer values. The authors then compare 6 variational neural MI estimation methods under a bilinear, separable, and joint critic setting. They have multiple novel findings including the joint critic exceeding the performance of other approaches on unstructured datasets, the lack of a clear winner through increasing critic capacity and critic method, and the differences in the impact of the parameters across Gaussian and unstructured settings.

---

> ### Author Rebuttal · Authors · 2024-08-16
>
> Thank you for the insightful comments. Our point-by-point answers to the reviewer’s concerns and questions are provided below.
>
> [Q1] **Non-critic approaches**
>    - We chose to exclude non-critic approaches, such as binning and non-parametric kernel-density estimators, due to their well-known limitations in scaling with sample size or dimensionality [5, 6, R1]. These limitations essentially make the non-critic approaches not applicable to text and image datasets, and the popularization of neural MI estimators, starting with MINE, was specifically to overcome these limitations. Neural estimators have demonstrated superior performance over non-critic methods in previous works (e.g., [20]), justifying our focus on neural estimators in our benchmark. We recognize the importance of clarifying this scope in the manuscript and will revise it accordingly. Thanks for your comment.
>    - [R1] S Gao et al., Efficient Estimation of Mutual Information for Strongly Dependent Variables, AISTATS, 2015.
>
> [Q2] **Nuisance for text datasets**
>    - Thank you for the comment. In our study, we introduced nuisance to image datasets to make them more realistic without affecting the true MI value, as simple synthetic images (e.g., plain 0/1 only images) may not be representative of practical image datasets (lines 165-166). In contrast, the text datasets we used, such as sentence embeddings from the IMDB dataset, already follow practical natural language distributions. Therefore, we did not introduce additional nuisances to the text data, as they are already realistic in this context.
>    - However, if we consider the formal definition of nuisance in Definition 4.3 (lines 126-128), it is indeed possible to add nuisance to any type of dataset. For example, one could concatenate unrelated inputs, such as song lyrics, to the text data used in our study. While this would maintain the consistency of MI, it would result in datasets that deviate from practical text distributions. For these reasons, our discussion on nuisance focuses on image datasets.
>    - We acknowledge that this discussion is beneficial for broadening understanding, and we will address the items mentioned above in the revised manuscript.
>
>
> [Q3] **Clarity of writing**
>    - Thank you for the feedback. To improve the clarity of the paper, we will move some of the plots to the supplementary materials. For instance, the results for the mixture of images (4th row) and possibly sentence embeddings (3rd row) in Figure 3, which may be somewhat redundant, will be moved to the supplementary materials.
>    - In addition, we will streamline the detailed explanations in the background section. Specifically, explanations of the variational bounds in lines 78-79 will be moved to the supplementary materials with further details.
>    - We will incorporate these revisions and other improvements based on your comments to make the paper easier to read and more accessible.
>
> [Q4] **Superiority of SMILE-5**
>    - Thank you for raising this point. While SMILE-5 performs well in Gaussian and NLP cases, as mentioned in the manuscript, it shows significantly lower performance than NWJ and MINE in vision cases with large true MI values. This variation in performance across different data domains led us to conclude that there is no universal winner among the estimators.

---

> > ### Comment · Reviewer_AxEX · 2024-08-22
> > **Response to rebuttal**
> >
> > Thanks for your response. I have decided to raise my score.

---

> > > ### Author Response · Authors · 2024-08-22
> > >
> > > Dear reviewer AxEX, thank you for the positive comment. - Authors

---

### Author Rebuttal · Authors · 2024-08-16

Dear Reviewers,

We would like to express our sincere gratitude to the reviewers for their thoughtful and constructive feedback on our manuscript. We greatly appreciate the time and effort each reviewer has taken to engage with our work, and we welcome further discussions that can help improve the quality of the manuscript.

Here, we summarize the strengths of our paper as highlighted in the reviews:

1. Novel contributions:
    - Development of a benchmark suite for evaluating neural MI estimators on unstructured data, providing a unified framework for future research (7BAF, GBdp).
    - Insight into scenarios where neural MI methods may accurately estimate true MI, reducing resource requirements by showing that critic capacity does not significantly impact performance (AxEX).
2. Methodology and theoretical foundation:
    - Well-motivated methods for constructing datasets with specific MI values, supported by theoretical proofs (AxEX, 7BAF).
    - Introduction of methods to control ground truth MI values in datasets, enabling robust evaluations (7BAF, GBdp).
3. Practical and empirical Insights:
    - Provision of prescriptive results, such as the robustness of MINE to noise, which can guide future research (AxEX).
    - Empirical results offer valuable insights into the performance of MI estimators in realistic scenarios (GBdp).
4.  Clarity and presentation:
    - The manuscript is well-written, easy to follow, and effectively communicates the motivation and findings (7BAF, GBdp).
    - The claims in the paper are correct and support the provided conclusions, with no major concerns regarding their correctness (AxEX, 7BAF, GBdp).

We thank the reviewers for their positive reviews, and we will provide additional responses to the specific questions and opportunities for improvements in our official comments for each reviewer.

---

### Decision · Program_Chairs · 2024-09-26

**Decision:**

Accept (Poster)

**Comment:**

The paper provides valuable insights into scenarios where Neural Mutual Information (MI) methods can accurately estimate the true MI. It also highlights that critic capacity does not significantly impact the performance of MI methods, which could help future researchers test their methods with lower resource requirements. Reviewers liked the paper and engaged with the authors during rebuttal. I concur and think this paper could be accepted if there is space.